# Longitudinal association between e-cigarette use and respiratory symptoms among US adults: Findings from the Population Assessment of Tobacco and Health Study Waves 4–5

Emma Karey[1], Shu Xu[2], Pan He[3], Raymond S. Niaura[2], Charles M. Cleland[3], Elizabeth R. Stevens[3], Scott E. Sherman[3,4], Omar El-Shahawy[2,3], Jennifer Cantrell[2], Nan Jiang[3]*

1 Department of Medicine, New York University Grossman School of Medicine, New York, NY, United States of America, 2 School of Global Public Health, New York University, New York, NY, United States of America, 3 Department of Population Health, New York University Grossman School of Medicine, New York, NY, United States of America, 4 Department of Medicine, VA New York Harbor Healthcare System, New York, NY, United States of America

* Nan.Jiang@nyulangone.org

**Data Availability Statement:** The Population Assessment of Tobacco and Health (PATH) Study

## Abstract

### Background

We assessed longitudinal effects of e-cigarette use on respiratory symptoms in a nationally representative sample of US adults by combustible tobacco smoking status.

### Methods

We analyzed Waves 4–5 public-use data from the Population Assessment of Tobacco and Health Study. Study sample included adult respondents who reported no diagnosis of respiratory diseases at Wave 4, and completed Waves 4–5 surveys with no missing data on analytic variables (N = 15,291). Outcome was a validated index of functionally important respiratory symptoms based on 7 wheezing/cough questions (range 0–9). An index score of ≥2 was defined as having important respiratory symptoms. Weighted lagged logistic regression models were performed to examine the association between e-cigarette use status at Wave 4 (former/current vs. never use) and important respiratory symptoms at Wave 5 by combustible tobacco smoking status (i.e., never/former/current smokers), adjusting for Wave 4 respiratory symptom index, sociodemographic characteristics, secondhand smoke exposure, body mass index, and chronic disease.

### Results

Among current combustible tobacco smokers, e-cigarette use was associated with increased odds of reporting important respiratory symptoms (former e-cigarette use: adjusted odds ratio [AOR] = 1.39, 95% confidence interval [CI]: 1.07–1.81; current e-

data are available at the website of the National Addiction & HIV Data Archive Program (https://www.icpsr.umich.edu/web/NAHDAP/studies/36498/datadocumentation).

**Funding:** This work was supported by the National Cancer Institute (NCI; R21CA260423), National Institute of Environmental Health Sciences (NIEHS; T32ES007324), and National Heart, Lung, and Blood Institute (NHLBI; R01HL139239) of the National Institutes of Health. The NCI provided support in the form of salaries for NJ, SX, PH, RSN, and CMC. The NIEHS and NHLBI provided support in the form of salaries for EK. The funders played no role in study design, data management and analysis, preparation of the manuscript, or decision to publish.

**Competing interests:** Dr. Emma Karey is an employee and owns stock in AstraZeneca. No pharmaceutical financial support was provided for this work and all contributions occurred outside of working hours. All remaining authors declared no competing interests.This does not alter our adherence to PLOS ONE policies on sharing data and materials.

cigarette use: AOR = 1.55, 95% CI: 1.17–2.06). Among former combustible tobacco smokers, former e-cigarette use (AOR = 1.51, 95% CI: 1.06–2.15)—but not current e-cigarette use (AOR = 1.59, 95% CI: 0.91–2.78)—was associated with increased odds of important respiratory symptoms. Among never combustible tobacco smokers, no significant association was detected between e-cigarette use and important respiratory symptoms (former e-cigarette use: AOR = 1.62, 95% CI: 0.76–3.46; current e-cigarette use: AOR = 0.82, 95% CI: 0.27–2.56).

## Conclusions

The association between e-cigarette use and respiratory symptoms varied by combustible tobacco smoking status. Current combustible tobacco smokers who use e-cigarettes have an elevated risk of respiratory impairments.

## Introduction

Combustible tobacco products (e.g., cigarettes) contain more than 7,000 toxic chemicals and cause cardiopulmonary morbidities and mortalities, including asthma, chronic bronchitis, and chronic obstructive pulmonary disease (COPD) [1]. E-cigarettes contain fewer toxicants than combustible tobacco [2]. But, several chemicals in e-cigarette aerosols are known pulmonary toxicants when inhaled (e.g., propylene glycol and diacetyl [3, 4]). Cross-sectional studies among adults suggest that e-cigarette use is associated with increased risks of respiratory diseases (e.g., asthma and COPD) [5–9] and/or respiratory symptoms (e.g., wheezing and cough) [9–11].

Longitudinal research surrounding the respiratory risk of e-cigarette use is limited [12–16]. Two studies used self-reported diagnosis of respiratory disease (e.g., COPD, chronic bronchitis, and asthma) as the outcome [12, 13]. Bhatta et al. [12] and Xie et al. [13] analyzed Waves 1–3 and Waves 1–4 data of the Population Assessment of Tobacco and Health (PATH) Study, respectively, and found that, among adults with no history of respiratory diseases at Wave 1, e-cigarette use (former/current vs. never use) was associated with increased odds of incident respiratory diseases at follow-up after adjusting for combustible tobacco smoking status. Findings suggested that e-cigarette use added additional respiratory risk independent of that from combustible tobacco smoking [12, 13].

Three longitudinal studies used self-reported respiratory symptoms as the outcome and found contradictory results [14–16]. Reddy et al. [14] analyzed Waves 3–4 PATH data and found that, among respondents with no wheezing or dry cough at Wave 3, e-cigarette use (current vs. non-current use) at Wave 3 was associated with increased odds of reporting wheezing or dry cough at Wave 4 among current tobacco smokers but not among non-current tobacco users. In contrast, Sargent et al. [15] analyzed Waves 2–3 PATH data and used a novel, validated index of functionally important respiratory symptoms (hereafter referred to as "respiratory symptom index") as the outcome. The study found that, among adults with no respiratory disease at Wave 2, e-cigarette use (current vs. non-current use) at Wave 2 was associated with worsening of respiratory health at Wave 3 (i.e., respiratory symptom index changing from <2 to ≥2 in Wave 2 through Wave 3) among non-cigarette smokers but not among current cigarette smokers [15]. Xie et al. [16] analyzed Waves 2–5 PATH data and found that, among young adults with no respiratory diseases or symptoms at baseline Waves, e-cigarette use

(former/current vs. never use) was associated with increased odds of incident respiratory symptoms, adjusting for combustible tobacco smoking status [16].

Given the inconsistent results from limited longitudinal studies, we examined the longitudinal effect of e-cigarette use on respiratory symptoms among US adults by combustible tobacco smoking status (i.e., never, former, and current smokers) using Waves 4–5 PATH data. Following previous studies [15–18], we used the respiratory symptom index as the outcome. Respiratory symptom index is derived based on seven PATH items that assess wheezing and cough [17].

## Materials and methods

### Study population

We used Waves 4–5 public-use data of the PATH Study, a nationally representative longitudinal cohort study on tobacco use and related health outcomes of US population aged ≥ 12 years. Data were collected between 2016–2018 (Wave 4) and 2018–2019 (Wave 5) [19]. Details about the PATH Study are available elsewhere [20]. As showed in Fig 1, respondents aged ≥18 years who completed both Waves 4 and 5 surveys, reported no diagnosis of any respiratory diseases (including COPD, chronic bronchitis, emphysema, asthma, and other lung or respiratory conditions) at Wave 4, and had no missing data on any analytic variables in both waves were

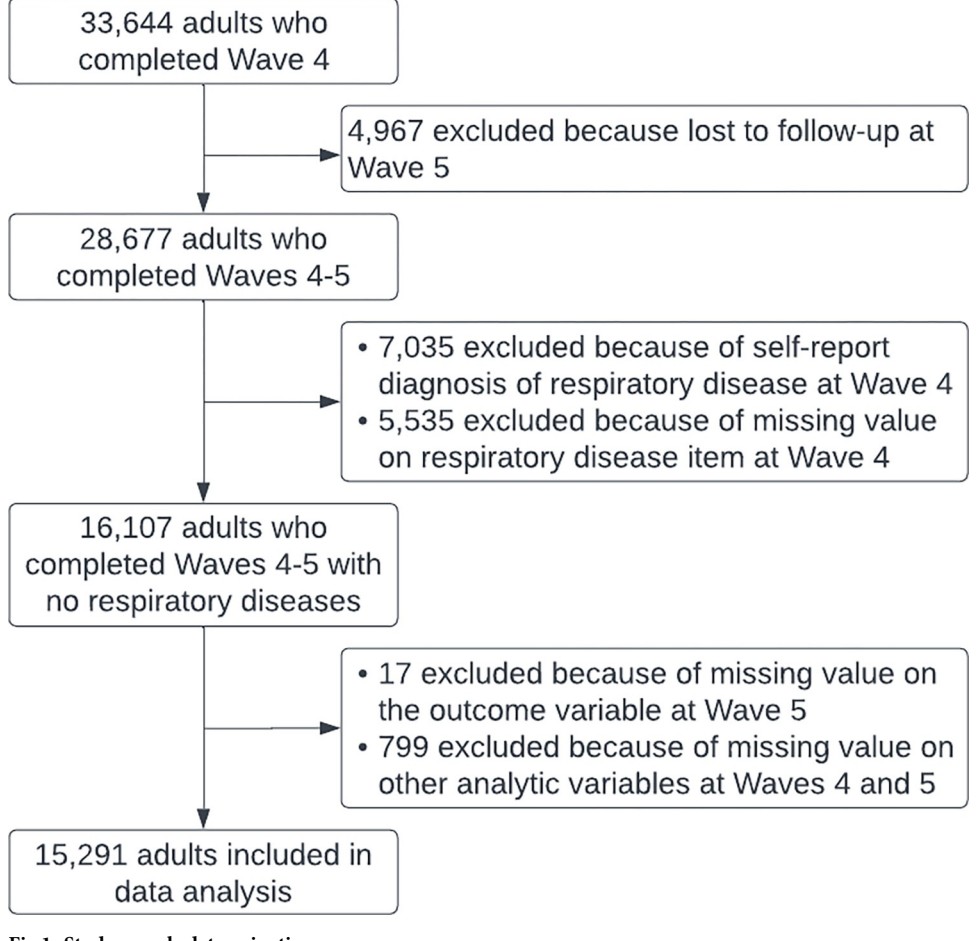

**Fig 1. Study sample determination.**

included in data analyses. Among 33,644 adult respondents of Wave 4, 15,291 (45.4%) met the above criteria and were included in our analyses. Ethical review and approval were waived for this study due to the public and de-identified nature of PATH public-use data.

## Key measures

**Respiratory symptom index.** Following prior research [15–18], we calculated the respiratory symptom index based on six wheezing items and a nighttime dry cough item (Table 1). With a range of 0–9, higher index values indicate more respiratory symptoms. When used as the outcome variable, it was coded as a binary variable and an index score of ≥2 was defined as having important respiratory symptoms [17].

**E-cigarette use and combustible tobacco smoking.** E-cigarette use was assessed by a series of questions. Responses were classified into three groups, including never, former, and current use. Never use was defined as never use of any electronic nicotine products. Former use was defined as ever use of electronic nicotine products but not in the past 30 days. Current use was defined as past 30-day use of any electronic nicotine products. Similarly, a series of questions assessed the use of combustible tobacco products, including cigarettes, traditional cigars, cigarillos, filtered cigars, pipes, and hookah. Respondents were classified as never smokers (never use of any combustible tobacco products), former smokers (ever use of any

**Table 1. Functionally important respiratory symptom index: Measures and calculation.**

| PATH questions | Response options | Point |
|---|---|---|
| Q1. Have you ever had wheezing or whistling in the chest at any time in the past? | Yes<br>No | 0 (Q1 'No')<br>1 (Q1 'Yes' AND Q2 'No') |
| Q2. Have you had wheezing or whistling in the chest in the past 12 months? | Yes<br>No | 1 (Q1 'Yes' AND Q2 'Yes' AND Q3 'None') |
| Q3. How many attacks of wheezing have you had in the past 12 months? | None<br>1–3<br>4–12<br>More than 12 | 2 (Q1 'Yes' AND Q2 'Yes' AND Q3 '1–3')<br>3 (Q1 'Yes' AND Q2 'Yes' AND Q3 '4–12')<br>4 (Q1 'Yes' AND Q2 'Yes' AND Q3 'More than 12') |
| Q4. In the past 12 months, how often, on average has your sleep been disturbed due to wheezing? | Never<br>< 1 night per week<br>≥1 night per week | 0<br>1<br>2 |
| Q5. In the past 12 months, has wheezing ever been severe enough to limit your speech to only one or two words between breaths? | Yes<br>No | 0<br>1 |
| Q6. In the past 12 months, has your chest sounded wheezy during or after exercise? | Yes<br>No | 0<br>1 |
| Q7. In the past 12 months, have you had a dry cough at night? | Yes<br>No | 0<br>1 |
| **Derived variable: Functionally important respiratory symptom index** | **Sum of the above points** (range: 0–9) | |
| **Derived outcome variable: Important respiratory symptoms** | **No** (if functionally important respiratory symptom index <2)<br>**Yes** (if functionally important respiratory symptom index ≥2) | |

combustible tobacco products but not in the past 30 days), or current smokers (past 30-day use of any combustible tobacco products).

**Covariates.** Covariates included age, gender (male/female), race/ethnicity (non-Hispanic White, non-Hispanic Black, Hispanic, and other), education (less than high school, high school equivalency diploma, high school graduate, some college, and Bachelor or advanced degree), and annual household income (<$50,000, ≥$50,000, and unreported). Following prior studies [6, 10–12], covariates also included body mass index (BMI; underweight, normal weight, overweight, and obese), secondhand smoke exposure ("During the past 7 days, about how many hours were you around others who were smoking? Include time in your home, in a car, at school, or outdoors" with a numeric response), and chronic disease. Chronic disease was assessed by three items ("Has a doctor, nurse or other health professional told you that you had any of the following heart conditions: high blood pressure, high cholesterol, congestive heart failure, stroke, heart attack, other heart condition? [Yes/No]" "Have you ever been told by a doctor, nurse or other health professional that you had cancer? [Yes/No]" "Have you ever been told by a doctor, nurse or other health professional that you had diabetes, sugar diabetes, high blood sugar, or borderline diabetes? [Yes/No]"). Respondents who reported "no" to all the three questions were considered having no chronic disease, and otherwise as having chronic disease(s).

## Statistical analysis

Data analyses were performed from January through July, 2022 using SAS version 9.4. PATH sample weights were used to account for the survey sampling frame, nonresponse, and selection bias. Descriptive statistics summarized Wave 4 sample characteristics. Rao-Scott chi-square tests (for categorical variables) and ANOVA (for continuous variables) were conducted to compare sample characteristics by combustible tobacco smoking status. Separate lagged logistic regression was performed to examine the association between Wave 4 e-cigarette use status (former/current vs. never) and important respiratory symptoms at Wave 5 by combustible tobacco smoking status, adjusting for Wave 4 respiratory symptom index (as a continuous variable) and covariates (i.e., sociodemographic characteristics, BMI, secondhand smoke exposure, and chronic disease).

## Results

Our sample included 53% females and 66.3% non-Hispanic White (Table 2). One-third (34.4%) had obtained bachelor's or advanced degree, and 44.9% had an annual household income of less than $50,000. On average, participants reported secondhand smoke exposure for 0.85 hours (SE = 0.03) in the past 7 days. More than half (53.4%) of participants reported a diagnosis of chronic disease. At Wave 4, 51.2% and 19.1% of participants reported former and current combustible tobacco smoking, respectively, and 17.1% and 5.1% reported former and current e-cigarette use, respectively. S1 Table shows the frequency of e-cigarette use and combustible cigarette smoking in the past 30 days at Wave 4. E-cigarette use was more prevalent among current and former combustible tobacco smokers than among never combustible tobacco smokers (Rao-Scott $\chi^2_{(4)}$ = 6669.70, $p$ < .001).

At Wave 4, the sample reported a mean respiratory symptom index of 0.50 (SE = 0.01). The index was higher among current combustible tobacco smokers (1.01, SE = 0.03) than former (0.41, SE = 0.02) and never combustible tobacco smokers (0.32, SE = 0.02; $F_{(2, 15288)}$ = 434.49, $p$ < .001). Overall, 10.1% of participants reported important respiratory symptoms at Wave 4 (data not reported in tables). Former and current combustible tobacco smokers were more likely to report important respiratory symptoms than never combustible tobacco smokers

**Table 2. Sample characteristics at Wave 4.**

| | All sample (n = 15291) | | Combustible tobacco smoking status | | | | | | Rao-Scott Chi-square or F-value |
|---|---|---|---|---|---|---|---|---|---|
| | | | Never (n = 3638; 29.7%) | | Former (n = 6380; 51.2%) | | Current (n = 5273; 19.1%) | | |
| | n | (%)[a] | n | (%)[a] | n | (%)[a] | n | (%)[a] | |
| Gender | | | | | | | | | 123.02* |
| Female | 8164 | (53.0) | 2204 | (62.5) | 3358 | (50.5) | 2602 | (45.0) | |
| Male | 7127 | (47.0) | 1434 | (37.5) | 3022 | (49.5) | 2671 | (55.0) | |
| Age (years) | | | | | | | | | 390.82* |
| 18–24 | 5271 | (12.5) | 2137 | (16.7) | 1625 | (7.8) | 1509 | (18.4) | |
| 25–34 | 2835 | (16.2) | 391 | (16.9) | 1279 | (13.2) | 1165 | (23.1) | |
| 35–44 | 1977 | (15.4) | 281 | (15.4) | 895 | (14.7) | 801 | (17.4) | |
| 45–54 | 1826 | (16.1) | 264 | (14.8) | 841 | (17.0) | 721 | (15.7) | |
| 55–64 | 1867 | (18.9) | 262 | (15.6) | 865 | (21.5) | 740 | (17.2) | |
| > 64 | 1515 | (20.9) | 303 | (20.6) | 875 | (25.8) | 337 | (8.2) | |
| Race/Ethnicity | | | | | | | | | 158.24* |
| Non-Hispanic White | 8999 | (66.3) | 1777 | (57.6) | 4056 | (72.1) | 3166 | (64.2) | |
| Non-Hispanic Black | 2253 | (11.1) | 652 | (13.1) | 729 | (8.3) | 872 | (15.4) | |
| Hispanic | 2861 | (14.1) | 884 | (17.4) | 1114 | (12.1) | 863 | (14.4) | |
| Other | 1178 | (8.5) | 325 | (11.9) | 481 | (7.5) | 372 | (6.0) | |
| Education level | | | | | | | | | 296.86* |
| Less than high school | 1584 | (8.4) | 486 | (10.5) | 402 | (5.9) | 696 | (11.5) | |
| GED | 716 | (3.8) | 90 | (2.6) | 194 | (2.9) | 432 | (8.1) | |
| High school graduate | 3513 | (22.3) | 1102 | (24.1) | 1098 | (19.5) | 1313 | (27.0) | |
| Some college | 5487 | (31.1) | 1262 | (29.9) | 2249 | (30.4) | 1976 | (35.1) | |
| Bachelor or advanced degree | 3991 | (34.4) | 698 | (32.9) | 2437 | (41.3) | 856 | (18.3) | |
| Annual household income | | | | | | | | | 186.58* |
| < $50,000 | 8068 | (44.9) | 1943 | (47.6) | 2783 | (37.8) | 3342 | (60.0) | |
| ≥ $50,000 | 6674 | (51.5) | 1481 | (47.9) | 3414 | (58.9) | 1779 | (37.0) | |
| Unreported | 549 | (3.6) | 214 | (4.5) | 183 | (3.3) | 152 | (3.0) | |
| Secondhand smoke exposure (hours), Mean (SE) | 0.85 | (0.03) | 0.28 | (0.03) | 0.44 | (0.04) | 2.83 | (0.12) | 803.13* |
| Body mass index | | | | | | | | | 28.47* |
| Underweight (<18.5) | 373 | (1.6) | 150 | (2.3) | 102 | (1.0) | 121 | (2.1) | |
| Normal Weight (18.5–24.9) | 5542 | (32.1) | 1487 | (33.4) | 2176 | (30.9) | 1879 | (33.0) | |
| Overweight (25–29.9) | 4742 | (34.1) | 1003 | (32.4) | 2093 | (35.2) | 1646 | (34.0) | |
| Obese (≥30) | 4634 | (32.2) | 998 | (31.9) | 2009 | (32.9) | 1627 | (30.9) | |
| Chronic disease | | | | | | | | | 70.97* |
| No | 9182 | (46.6) | 2602 | (52.1) | 3546 | (41.2) | 3034 | (52.2) | |
| Yes | 6109 | (53.4) | 1036 | (47.9) | 2834 | (58.8) | 2239 | (47.8) | |
| Respiratory symptom index, Mean (SE) | 0.50 | (0.01) | 0.32 | (0.02) | 0.41 | (0.02) | 1.01 | (0.03) | 434.49* |
| E-cigarette use | | | | | | | | | 6669.70* |
| Never use | 8808 | (77.8) | 3323 | (97.4) | 4016 | (83.2) | 1469 | (32.6) | |
| Former use | 4855 | (17.1) | 250 | (2.2) | 1934 | (13.9) | 2671 | (48.7) | |
| Current use | 1628 | (5.1) | 65 | (0.4) | 430 | (2.9) | 1133 | (18.7) | |

*Notes.* GED: general equivalency diploma

[a]Weighted percentages

*$p < .001$

(Rao-Scott $\chi^2_{(2)}$ = 320.03, $p < .001$). At Wave 5, the sample reported an average respiratory symptom index of 0.54 (SE = 0.01) and 11.1% reported important respiratory symptoms (data not reported). Former and current combustible tobacco smokers were more likely to report important respiratory symptoms than never combustible tobacco smokers (Rao-Scott $\chi^2_{(2)}$ = 271.30, $p < .001$).

Among never combustible tobacco smokers, e-cigarette use (vs. never use) at Wave 4 was not associated with increased odds of reporting important respiratory symptoms at Wave 5 (former e-cigarette use: adjusted odds ratio (AOR) = 1.62, 95% confidence interval (CI): 0.76–3.46, $p = .205$; current e-cigarette use: AOR = 0.82, 95% CI: 0.27–2.56, $p = .736$; Table 3). Among former combustible tobacco smokers, former e-cigarette use (AOR = 1.51, 95% CI: 1.06–2.15, $p = .024$) but not current e-cigarette use (AOR = 1.59, 95% CI: 0.91–2.78, $p = .102$) was associated with a higher odd of important respiratory symptoms. Among current combustible tobacco smokers, e-cigarette use was associated with higher odds of reporting important respiratory symptoms (former e-cigarette use: AOR = 1.39, 95% CI: 1.07–1.81, $p = .015$; current e-cigarette use: AOR = 1.55, 95% CI: 1.17–2.06, $p = .003$).

## Discussion

This population-based study contributed new evidence regarding the longitudinal effect of e-cigarette use on respiratory symptoms in a large nationally representative sample of US adults. Among current combustible tobacco smokers with no diagnosis of respiratory diseases at baseline, the combined use of e-cigarettes increased the adjusted odds of reporting important respiratory symptoms in a two-year period. The findings suggest that e-cigarette use increases the risk of developing important respiratory symptoms for current combustible tobacco smokers. In other words, current combustible tobacco smokers who have ever used e-cigarettes (former or current use) may be at elevated risks for developing respiratory symptoms than those who exclusively smoke combustible tobacco. Data from previous longitudinal studies are conflicting, with some finding e-cigarette use increases respiratory risks among current tobacco smokers [14] while others found no association between e-cigarette use and respiratory symptoms among current combustible cigarette smokers [16]. The inconsistent findings may be related to different study samples: The present study focused on adults aged ≥ 18 years whereas in Xie et al.'s study, only young adults (aged 18–24 years) were included in the analyses [16]. Future research investigating the effects of e-cigarettes on respiratory health may consider to include a stratification variable of age.

Among former combustible tobacco users, adjusted odds of reporting important respiratory symptoms depended on e-cigarette use status. Former e-cigarette use—but not current use—was found to be associated with higher odds of reporting important respiratory symptoms. Considering the complex trajectory and patterns of tobacco use, former combustible tobacco smokers who formerly used e-cigarettes may indicate a history of concurrent use of both tobacco products at the same time or using one tobacco product at a time and then switching to a different one, or a combined scenario. Similarly, former combustible tobacco smokers who currently used e-cigarettes may indicate complete switching from combustible tobacco to e-cigarettes or picking up e-cigarettes after quitting combustible tobacco. Future research needs to take into account the history of tobacco use when assessing health risks associated with e-cigarette use.

Consistent with a previous longitudinal epidemiological study [14], we detected no association between e-cigarette use and important respiratory symptoms among never combustible tobacco smokers. But, two previous studies analyzing PATH data found that, for never combustible cigarette smokers, e-cigarette use is associated with incident respiratory symptoms

**Table 3. Longitudinal association of e-cigarette use with important respiratory symptoms by combustible tobacco smoking status.**

| | Among never combustible tobacco smokers | | | Among former combustible tobacco smokers | | | Among current combustible tobacco smokers | | |
|---|---|---|---|---|---|---|---|---|---|
| | AOR | [95% CI] | *P* | AOR | [95% CI] | *P* | AOR | [95% CI] | *P* |
| E-cigarette use | | | | | | | | | |
| Never use | 1.00 | | | 1.00 | | | 1.00 | | |
| Former use | 1.62 | [0.76, 3.46] | .205 | 1.51 | [1.06, 2.15] | .024 | 1.39 | [1.07, 1.81] | .015 |
| Current use | 0.82 | [0.27, 2.56] | .736 | 1.59 | [0.91, 2.78] | .102 | 1.55 | [1.17, 2.06] | .003 |
| Gender | | | | | | | | | |
| Female | 1.00 | | | 1.00 | | | 1.00 | | |
| Male | 1.19 | [0.70, 2.03] | .523 | 0.82 | [0.63, 1.06] | .130 | 0.88 | [0.74, 1.05] | .155 |
| Age (years) | | | | | | | | | |
| 18–24 | 1.00 | | | 1.00 | | | 1.00 | | |
| 25–34 | 0.96 | [0.37, 2.45] | .927 | 1.20 | [0.80, 1.79] | .369 | 1.25 | [0.95, 1.65] | .112 |
| 35–44 | 0.80 | [0.34, 1.87] | .604 | 0.89 | [0.57, 1.40] | .611 | 1.33 | [1.03, 1.73] | .031 |
| 45–54 | 1.09 | [0.45, 2.63] | .851 | 1.24 | [0.72, 2.12] | .436 | 1.36 | [0.99, 1.88] | .056 |
| 55–64 | 0.73 | [0.31, 1.69] | .458 | 1.16 | [0.66, 2.03] | .608 | 1.52 | [1.11, 2.07] | .009 |
| > 64 | 1.15 | [0.53, 2.48] | .722 | 2.14 | [1.23, 3.71] | .008 | 1.18 | [0.78, 1.78] | .438 |
| Race/Ethnicity | | | | | | | | | |
| Non-Hispanic White | 1.00 | | | 1.00 | | | 1.00 | | |
| Non-Hispanic Black | 0.80 | [0.41, 1.58] | .523 | 0.97 | [0.67, 1.41] | .868 | 0.78 | [0.61, 1.00] | .051 |
| Hispanic | 0.52 | [0.21, 1.33] | .171 | 0.84 | [0.48, 1.46] | .533 | 0.75 | [0.50, 1.13] | .165 |
| Other | 1.04 | [0.39, 2.82] | .934 | 1.10 | [0.58, 2.08] | .776 | 1.31 | [0.86, 2.00] | .208 |
| Education level | | | | | | | | | |
| Less than high school | 0.65 | [0.26, 1.62] | .353 | 0.92 | [0.47, 1.80] | .804 | 1.01 | [0.75, 1.36] | .947 |
| GED | 1.07 | [0.25, 4.64] | .926 | 2.08 | [1.08, 4.03] | .030 | 0.81 | [0.58, 1.12] | .199 |
| High school graduate | 0.96 | [0.45, 2.06] | .913 | 1.11 | [0.75, 1.65] | .585 | 0.97 | [0.76, 1.25] | .815 |
| Some college | 1.00 | | | 1.00 | | | 1.00 | | |
| Bachelor or advanced degree | 0.94 | [0.45, 1.97] | .868 | 0.77 | [0.53, 1.14] | .188 | 0.64 | [0.49, 0.85] | .002 |
| Annual household income | | | | | | | | | |
| < $50,000 | 1.08 | [0.65, 1.81] | .756 | 0.87 | [0.64, 1.18] | .357 | 0.91 | [0.71, 1.17] | .454 |
| ≥ $50,000 | 1.00 | | | 1.00 | | | 1.00 | | |
| Unreported | 0.49 | [0.16, 1.46] | .197 | 1.61 | [0.65, 4.00] | .299 | 0.68 | [0.35, 1.31] | .245 |
| Secondhand smoke exposure | 1.02 | [0.93, 1.13] | .652 | 1.04 | [0.98, 1.10] | .176 | 1.02 | [1.00, 1.04] | .020 |
| Body mass index | | | | | | | | | |
| Underweight (<18.5) | 0.63 | [0.15, 2.69] | .528 | 0.98 | [0.16, 6.10] | .985 | 1.06 | [0.58, 1.94] | .855 |
| Normal Weight (18.5–24.9) | 1.00 | | | 1.00 | | | 1.00 | | |
| Overweight (25–29.9) | 2.04 | [1.13, 3.67] | .019 | 1.89 | [1.22, 2.92] | .005 | 0.98 | [0.78, 1.22] | .841 |
| Obese (≥30) | 2.18 | [1.14, 4.15] | .018 | 2.75 | [1.80, 4.22] | < .001 | 1.13 | [0.89, 1.44] | .298 |
| Chronic disease | | | | | | | | | |
| No | 1.00 | | | 1.00 | | | 1.00 | | |
| Yes | 0.85 | [0.49, 1.49] | .569 | 0.95 | [0.66, 1.35] | 0.754 | 1.10 | [0.88, 1.39] | .401 |
| Respiratory symptom index at Wave 4 | 2.81 | [2.10, 3.75] | < .001 | 2.83 | [2.29, 3.50] | < .001 | 2.08 | [1.92, 2.25] | < .001 |

*Notes.* AOR: adjusted odds ratio; CI: confidence interval; GED: general equivalency diploma.

among young adults [16] or worsening of respiratory symptoms among adults [15], suggesting that exclusive e-cigarette use may pose respiratory risks. More research is warranted to investigate respiratory effects of e-cigarette use among never combustible tobacco smokers.

Understanding the association of e-cigarette use and respiratory health has important implications for regulatory efforts and intervention programs that address e-cigarette use. In 2020, 5.1% of US adults reported current e-cigarette use [21]. E-cigarette use is more prevalent among current (13.8%) and former cigarette smokers (13.2%) than among never cigarette smokes (2.3%) [21]. Combustible cigarette smokers commonly cite a desire to quit or reduce cigarette smoking as a reason for concurrent e-cigarette use [22–24]. A recent Cochrane systematic review concluded that nicotine-containing e-cigarettes increased smoking abstinence rates compared to no e-cigarette use [25]. However, cigarette smokers should be informed that they must switch completely to achieve the health benefit, as higher levels of nicotine and toxicant (e.g., tobacco-specific nitrosamines, volatile organic compounds, and metals) have been detected among dual users than among cigarette-only smokers [26], indicating additional health risks associated with combined e-cigarette use [27].

This study has several limitations. First, public-use PATH data files contain no information about years of combustible tobacco smoking or years of being abstinent from combustible tobacco, which may impact respiratory health. There may be other variables (e.g., marijuana use) affecting the outcome. Future research is warranted to replicate the analyses accounting for those variables (e.g., cigarette pack years and marijuana use [15]). Second, current e-cigarette users included occasional users and regular users. Similarly, current combustible tobacco smokers included light smokers and heavy smokers, as well as individuals who smoked single combustible tobacco product and those who smoked multiple tobacco products. Future studies are needed to investigate whether these patterns have any impact on respiratory health conditions. Moreover, self-reported data are subject to recall bias and reporting errors. Despite the limitations, strengths of this study include the use of data collected from a nationally representative sample of adults and use of a validated index of a compact measure of functionally important respiratory symptoms as the outcome which captures early symptoms indicative of pulmonary impairment.

## Conclusions

This study contributes new evidence on the risk profile of e-cigarettes and is among the few studies that investigate longitudinal impacts of e-cigarette use on respiratory health. Educational interventions need to inform current combustible tobacco smokers about the additional risks of respiratory impairment derived from dual use of e-cigarettes and combustible tobacco product(s). Future research needs to focus on exclusive e-cigarette users who are naïve to combustible tobacco smoking (mostly youth and young adults) or former combustible tobacco smokers to examine if e-cigarette use causes respiratory symptoms and/or diseases.

## Supporting information

**S1 Table. Number of days using e-cigarettes and smoking combustible cigarettes in the past 30 days at Wave 4.**
(DOCX)

## Author Contributions

**Conceptualization:** Shu Xu, Nan Jiang.

**Formal analysis:** Shu Xu, Pan He.

**Funding acquisition:** Raymond S. Niaura, Nan Jiang.

**Methodology:** Shu Xu, Raymond S. Niaura, Charles M. Cleland, Nan Jiang.

**Project administration:** Nan Jiang.

**Supervision:** Shu Xu, Nan Jiang.

**Writing – original draft:** Emma Karey, Shu Xu, Nan Jiang.

**Writing – review & editing:** Emma Karey, Shu Xu, Pan He, Raymond S. Niaura, Charles M. Cleland, Elizabeth R. Stevens, Scott E. Sherman, Omar El-Shahawy, Jennifer Cantrell, Nan Jiang.

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
