## [Decision Letter · Decision Letter 0]

27 Sep 2023

PONE-D-23-19648Longitudinal association between e-cigarette use and respiratory symptoms among US adults: Findings from the Population Assessment of Tobacco and Health Study Waves 4 – 5PLOS ONE

Dear Dr. Jiang,

Thank you for submitting your manuscript to PLOS ONE. After careful consideration, we feel that it has merit but does not fully meet PLOS ONE’s publication criteria as it currently stands. Therefore, we invite you to submit a revised version of the manuscript that addresses the points raised during the review process.

We look forward to receiving your revised manuscript.

Kind regards,

Anindita Dutta, Ph.D.

Academic Editor

PLOS ONE

Journal Requirements:

2. 1) For studies reporting research involving human participants, PLOS ONE requires authors to confirm that this specific study was reviewed and approved by an institutional review board (ethics committee) before the study began. Please provide the specific name of the ethics committee/IRB that approved your study, or explain why you did not seek approval in this case.

  2) Please provide additional details regarding participant consent. In the ethics statement in the Methods and online submission information, please ensure that you have specified (1) whether consent was informed and (2) what type you obtained (for instance, written or verbal, and if verbal, how it was documented and witnessed). If your study included minors, state whether you obtained consent from parents or guardians. If the need for consent was waived by the ethics committee, please include this information.

"Dr. Emma Karey is an employee and owns stock in AstraZeneca. No pharmaceutical financial support was provided for this work and all contributions occurred outside of working hours. All remaining authors declared no competing interests."

We note that one or more of the authors are employed by a commercial company: AstraZeneca

2) Please also provide an updated Competing Interests Statement declaring this commercial affiliation along with any other relevant declarations relating to employment, consultancy, patents, products in development, or marketed products, etc.  

Within your Competing Interests Statement, please confirm that this commercial affiliation does not alter your adherence to all PLOS ONE policies on sharing data and materials by including the following statement: ""This does not alter our adherence to  PLOS ONE policies on sharing data and materials.” (as detailed online in our guide for authors http://journals.plos.org/plosone/s/competing-interests).

If this adherence statement is not accurate and  there are restrictions on sharing of data and/or materials, please state these. Please note that we cannot proceed with consideration of your article until this information has been declared.

5. We note that you have included the phrase “data not reported” in your manuscript. Unfortunately, this does not meet our data sharing requirements. PLOS does not permit references to inaccessible data. We require that authors provide all relevant data within the paper, Supporting Information files, or in an acceptable, public repository. Please add a citation to support this phrase or upload the data that corresponds with these findings to a stable repository (such as Figshare or Dryad) and provide and URLs, DOIs, or accession numbers that may be used to access these data. Or, if the data are not a core part of the research being presented in your study, we ask that you remove the phrase that refers to these data.

Reviewers' comments:

Reviewer's Responses to Questions

**Comments to the Author**

1. Is the manuscript technically sound, and do the data support the conclusions?

Reviewer #1: Partly

Reviewer #2: Yes

2. Has the statistical analysis been performed appropriately and rigorously? 

Reviewer #1: Yes

Reviewer #2: Yes

3. Have the authors made all data underlying the findings in their manuscript fully available?

Reviewer #1: Yes

Reviewer #2: Yes

4. Is the manuscript presented in an intelligible fashion and written in standard English?

Reviewer #1: Yes

Reviewer #2: Yes

5. Review Comments to the Author

Reviewer #1: This article is an interesting examination of the respiratory symptoms experience by dual users of e-cigarettes and combustible tobacco products using wave 4 and 5 of the PATH study. While I think this article is relevant and is an important addition to the literature, I do think a major revision is necessary to clarify the product user groups, study methods, and analyses. Combustible tobacco use and more specifically, combustible cigarettes pose serious risks to human health. While e-cigarettes are not without harm, there is well documented support that e-cigarette aerosol contains fewer numbers and lower levels of toxicants than smoke from combustible cigarettes. Incorporating or at least reporting the past 30-day frequencies and product use category frequencies (especially by combustible product category), is needed. Another factor that is likely important in understanding the respiratory risk of dual current combustible tobacco use and e-cigarette use that missing is the number of combustible products used in conjunction with e-cigarette use status. I understand all of these variables cannot always be included in models. However, I would at least consider reporting frequency of use as a table, and a note in the limitation section (e.g. unable to examine the frequency of past 30-day use or incorporate the number of combustible products used in the analyses due to sample size). If the article is examining the risk of dual use of e-cigarettes with combustible products, a bit of restructuring would help improve the clarity of this message to the reader.

Title:

- I think the authors should consider adjusting the title to clarify that this is the examination of e-cigarettes and combustible tobacco dual use(?) or at least have combustible tobacco in the title.

Abstract:

• I think in line 34, if “combustible tobacco smokers” was placed after former (to read as: “Among current and former combustible tobacco smokers and current e-cigarette..” instead of “Among current combustible tobacco smokers, former, and current e-cigarette..” Or maybe I am confused at the specific user groups referenced here? Please clarify in the methods the specific user groups to better understand the relationships examined.

• I would also clarify in the results section what your reference group is (i.e., never users? Current dual users vs. exclusive e-cigarette users?) when discussing the findings.

Introduction:

- Citation 14 found that dual use of e-cigarettes and cigarettes was associated with respiratory symptoms compared to those who used e-cigarettes exclusively or were a non-current user of either product. Please review for clarity/accuracy.

- The sentences following the reference to citation 16 (lines 70-71, i.e., “Such association is observed among never combustible cigarette smokers but not among current combustible cigarette smokers.”) and the following sentence (lines 71-72, i.e., “The results suggested that e-cigarettes pose respiratory risks, but add no additional risk independent of that from cigarette smoking”) are unclear. Please clarify what is mean by such association and review reference 16 for accuracy. I believe the authors only controlled for combustible use statistically and did not make specific comparisons to those who were former or current combustible tobacco users, as such I am not sure "but add no additional risk independent of that from cigarette smoking" means in this context.

- I would suggest that the authors give some context to what the respiratory symptom index is, what are “functionally important respiratory symptoms,” and what an increase in the reference index means (e.g., are higher scores related to more symptoms? Worse respiratory function?). Further, the authors write that current e-cigarette use was associated with worsening respiratory symptoms, but in the article referenced (article #15), those authors found that there was largely no increased symptom risk for exclusive use of several products including e-cigarettes (after adjustment for pack-years and marijuana) and that the risk ratio for several products including e-cigarettes was significantly lower compared to exclusive use of cigarettes. Maybe the context is missing for the risk associated with worsening respiratory symptoms? Please review and/or clarify.

Methods:

- In the Study Population section, it would be helpful to clarify if e-cigarette users needed to be a dual combustible product user to be included in your final sample. It is unclear currently how the inclusion/exclusion criteria are listed if exclusive e-cigarette users were excluded from the analytic sample.

- For the respiratory symptom index, I would review the references cited to make sure you are accurately reflecting those who used/categorized respiratory outcomes similarly or different. For example, citation 15 seems to have used this index and maybe the exact same way as suggested from introduction, but I am not sure. Please review citations for accuracy.

- I think it would be helpful to readers to present a bit more information about how the symptom index is calculated (e.g., are the yes/no responses counted toward the value of 9? Is this just an average of the frequency of episodes across all variables?). Maybe that would provide more clarity on what the cut-off value is indicative of (2 more symptoms in the past 12 months?). Additionally, I would clarify the dichotomization of the index (from a continuous score) for the use in the logistic regression analyses.

- In the e-cigarette and combustible smoking section, I would suggest some changes in how the user groups are labeled and a bit more clarity in how each group was created. I think “Ever Use” is a more accurate category, based on the limited information presented rather than “former use.” For example, from a questionnaire item like “Have you ever used an e-cigarette product with nicotine?” (yes/no) you can’t assume that the individual became a frequent user of that product and quit (like the term “former” suggests). Similarly, in the cigarette literature, a former smoker is typically defined as having smoked more than 100 cigarettes in their lifetime. I don’t think having ever smoked (even a puff) of a cigarette equates to being categorized as a former smoker. I would also encourage the authors to consider using a cut off for the number of days in the past 30-day used to define current use (see for example for using 5 or greater days in the past-30 days: Amato, M. S., Boyle, R. G., & Levy, D. (2016). How to define e-cigarette prevalence? Finding clues in the use frequency distribution. Tobacco control, 25(e1), e24-e29.).

- I might consider using “dual users of e-cigarettes and combustible tobacco products” to help clarify the user groups to the reader.

Results:

- I think a table is needed which outlines the frequencies of and past 30-day (number of days) product use by study wave. It would be helpful to understand the quantity and frequency of use as it relates to the respiratory index.

- In general, I would clarify when the results are referencing the entire sample (including never users) and when the results reference each product use category. For example, on page 16, line 162, and lines 168-169 is this among all participants?

- The results are hard to interpret without understanding the user groups more clearly. For example, lines 163-166 and 166-168 indicate significant differences in the respiratory status by former and current combustible tobacco use status, but do these results include dual users of e-cigarettes? Or are they combustible product users only?

- I think a key piece missing from the results are never e-cigarette users x combustible tobacco use status analyses. I believe analyses only examine using e-cigarettes x combustible tobacco use status (never, “former,” and current) but I did not see any analyses examining combustible tobacco use status x never e-cigarette use and how that compare to the outcomes presented currently. I think it is important to understand the respiratory risk comparisons between that user group (never e-cigarette vs. ever, and current combustible use) and the e-cigarette “dual” user groups (e-cigarette ever, current vs. ever, current combustible).

Discussion:

- Overall, thinking about a potential harm reduction strategy that includes e-cigarettes, I think that saying e-cigarettes present a risk to developing independent of combustible tobacco is relevant (and important!). However, I think the relativity of this risk (of e-cigarette exclusive use compared to combustible tobacco/cigarettes compared to dual use of both types of products) is a key piece missing to balance the discussion. This might help clarify that the message (I think!) about the unique respiratory risk associated with dual use patterns of combustible tobacco and e-cigarette use.

Reviewer #2: This Ms uses data from two waves of the PATH study to determine the relation between use of e-cigarettes and (apparently) onset of wheezing symptoms. Strengths of the study are in data from a representative sample, tests of longitudinal effects, and ability to control for several demographic covariates and (in some way) for prior respiratory symptoms. The authors studied both current and former e-cigarette use and conducted stratified analyses subgrouped on combustible cigarette status, finding significant effects among both current and former smokers. The results may have implications for informing current discussions about the health risks of e-cigarette use.

I found strengths in the study but there were several issues that seemed unclear. For example, the introduction around line 54 says that previous results were inconsistent. But when I looked at these studies and ones subsequently discussed I found that they all showed significant associations of at least one e-cigarette index with some types of respiratory symptoms. This doesn’t seem inconsistent to me.

I had difficulty understanding the criterion variable. First, the criterion variable is described in the abstract and text as “important respiratory symptoms.” However the measure employed is essentially an index of wheezing when in fact there are many other respiratory symptoms reflecting important health conditions (e.g., asthma, bronchitis) so I thought the language was misleading. Further it is sometimes said that the index was based on “core wheezing symptoms” but there was no discussion of what non-core wheezing symptoms might be.

The second point of ambiguity for me was whether the criterion measure was binary or continuous. The method section says that reports of symptoms were obtained with nine items and these were combined in an index. Ok, But at other points the continuous score was apparently dichotomized (>2 symptoms vs. <2 symptoms) and the continuous information discarded. Why? And where did this cutpoint come from. Without any justification it seems arbitrary. In the Results section, material at the top of page 10 seems to be dealing with a continuous criterion variable, but the material at page 10 bottom seems to be describing an incidence analysis with a binary criterion variable.

The third ambiguity is how the analyses were conducted. It is said that the analytic sample was persons who had no respiratory disease at baseline and that logistic regression analyses tested for prediction of a binary criterion variable at follow-up. Ok. But at other points it is said that the analyses controlled for prior respiratory symptoms, suggesting a continuous analysis. And Table 4 is mysterious because it does not say whether the criterion variable was dichotomous or continuous, and what appears to be a continuous covariate (“Respiratory symptom index”) is listed as one of the variables in the model, so I was unclear on what happened. Table 4 implies but does not clarify whether all the predictors were entered simultaneously. Thus I ended up being uncertain what was actually done and whether this was a true incidence analysis or some variant.

The discussion was respectable but left out the fact that a number of studies have found significant associations of e-cigarette use with respiratory disease among nonsmokers. This does not negate the conclusions of the present Ms but I think more context could be helpful.

In summary, I think this was basically a sound study and there are replicated results with health implications in there somewhere. But I found the presentation confusing at points and was left uncertain about whether this was an incidence analysis or some kind of onset analysis that controlled for baseline symptom levels.

Specific comments:

The abstract is variable on reporting accuracy. Some portions are correct but the summary leaves out the fact that elevated risks of e-cigarette use were found among former smokers as well as among current smokers.

line 50: Propylene glycol is a toxicant? It is one of the primary humectants in current e-cigarettes so I’m not sure where this is coming from. Also it is said in the discussion that nicotine is a toxicant but I think there are people who would disagree with this.

The present analyses covered a wide age range, going from 18 years to over 65 years. However, e-cigarette use is uncommon among older adults. Did the authors do sensitivity analyses testing for interactions with age? If not, why not.

6. PLOS authors have the option to publish the peer review history of their article (what does this mean?). If published, this will include your full peer review and any attached files.

Reviewer #1: No

Reviewer #2: No

---

## [Author Response · Author response to Decision Letter 0]

1 Dec 2023

We appreciate the time the reviewers have taken to review our work. We have attached detailed responses to all reviews’ comments and made changes in the revised manuscript with tracking changes.

---

## [Decision Letter · Decision Letter 1]

9 Feb 2024

PONE-D-23-19648R1Longitudinal association between e-cigarette use and respiratory symptoms among US adults: Findings from the Population Assessment of Tobacco and Health Study Waves 4 – 5PLOS ONE

Dear Dr. Jiang,

Thank you for submitting your manuscript to PLOS ONE. After careful consideration, we feel that it has merit but does not fully meet PLOS ONE’s publication criteria as it currently stands. Therefore, we invite you to submit a minorly revised version of the manuscript that addresses the points raised during the review process.

We look forward to receiving your revised manuscript.

Kind regards,

Sadaf G. Sepanlou, MD/MPH

Academic Editor

PLOS ONE

Journal Requirements:

**Additional Editor Comments:**

Please respond to comments raised by reviewer #2 and specifically comments number 4 and 5.

Reviewers' comments:

Reviewer's Responses to Questions

**Comments to the Author**

1. If the authors have adequately addressed your comments raised in a previous round of review and you feel that this manuscript is now acceptable for publication, you may indicate that here to bypass the “Comments to the Author” section, enter your conflict of interest statement in the “Confidential to Editor” section, and submit your "Accept" recommendation.

Reviewer #1: All comments have been addressed

Reviewer #2: (No Response)

2. Is the manuscript technically sound, and do the data support the conclusions?

Reviewer #1: Yes

Reviewer #2: Partly

3. Has the statistical analysis been performed appropriately and rigorously? 

Reviewer #1: Yes

Reviewer #2: Yes

4. Have the authors made all data underlying the findings in their manuscript fully available?

Reviewer #1: Yes

Reviewer #2: Yes

5. Is the manuscript presented in an intelligible fashion and written in standard English?

Reviewer #1: Yes

Reviewer #2: Yes

6. Review Comments to the Author

Reviewer #1: The authors have adequately addressed my comments and I appreciate their thoughtful responses. I have no further comments.

Reviewer #2: This is a revision of a Ms previously submitted to the journal. I thought that the authors responses to reviewer comments were generally appropriate though I felt that two responses tended to dodge the issue and one issue was ignored entirely.

My comments in order of significance (highest first):

1. The discussion says the present results are consistent with other studies in finding no association of e-cigarette use with respiratory symptoms among never smokers. That is not correct. There are in fact a number of studies that find exactly the opposite. One difference is that the other studies used actual diagnoses of disease (asthma or COPD) as the criterion. The authors say here that they are studying important respiratory symptoms but ignore research that has studied diagnoses of actual respiratory disease, as if the authors’ approach is the only way to do business and asthma and COPD are unimportant. For that it’s worth, I don’t think this is a correct approach. I think that in discussion, readers should be informed about evidence from all relevant studies. The authors may chose to pass over this comment but that is my view.

2. Definition of variables: 2a. The definition of current smoker apparently is not current exclusive smoker; this category includes poly users. Unless I missed something, that is not exclusive cigarette smoking because it includes individuals who used multiple tobacco products.. This needs to be made very clear. 2b. Similarly for e-cigarette use, is this really e-cigarette use exclusive of smoking? And of any other tobacco products? And what about marijuana? This is a significant issue because in many samples polyuse is the norm, and readers need to know exactly how the categories were defined. I don't know if the authors should be asked to reanalyze their data but I think this issue needs to highlighted in a revision.

3. The presentation of analyses and results is still confusing. When the authors say an analysis is on “important respiratory symptoms,” apparently this means a binary criterion variable. At other points, respiratory symptoms means a continuous index of symptoms. The Ms goes back and forth without justifying why one of these is important and the other isn’t. Apparently they think this is intuitively obvious but it isn't obvious to me.

4. It is still unclear to me how the authors can say that their longitudinal analyses are based on participants without any respiratory disease at baseline, and then report the mean number of respiratory symptoms they had at baseline Maybe I’m dense (I’ve been accused of worse) but it’s difficult for me to understand how you can have it both ways.

5. The authors say that coughing is a measure of chronic airway obstruction as in asthma. I believe that coughing can also be attributable to infectious causes, i.e., upper respiratory infections such as colds or flu.

7. PLOS authors have the option to publish the peer review history of their article (what does this mean?). If published, this will include your full peer review and any attached files.

Reviewer #1: No

Reviewer #2: No

---

## [Author Response · Author response to Decision Letter 1]

12 Feb 2024

Additional Editor Comments

COMMENT: Please respond to comments raised by reviewer #2 and specifically comments number 4 and 5.

RESPONSE: We appreciate the time that the editor and reviewers have taken to review our work. We have revised our paper accordingly. Please see details below.

Comments of Reviewer #1

COMMENT: The authors have adequately addressed my comments and I appreciate their thoughtful responses. I have no further comments.

RESPONSE: We are grateful for the reviewer’s comment.

Comments of Reviewer #2 

COMMENT: 1. The discussion says the present results are consistent with other studies in finding no association of e-cigarette use with respiratory symptoms among never smokers. That is not correct. There are in fact a number of studies that find exactly the opposite. One difference is that the other studies used actual diagnoses of disease (asthma or COPD) as the criterion. The authors say here that they are studying important respiratory symptoms but ignore research that has studied diagnoses of actual respiratory disease, as if the authors’ approach is the only way to do business and asthma and COPD are unimportant. For that it’s worth, I don’t think this is a correct approach. I think that in discussion, readers should be informed about evidence from all relevant studies. The authors may choose to pass over this comment but that is my view.

RESPONSE: We have revised the sentences, which now read as “Consistent with a previous longitudinal epidemiological study [14], we detected no association between e-cigarette use and important respiratory symptoms among never combustible tobacco smokers. But, two previous studies analyzing PATH data found that, for never combustible cigarette smokers, e-cigarette use is associated with incident respiratory symptoms among young adults [16] or worsening of respiratory symptoms among adults [15], suggesting that exclusive e-cigarette use may pose respiratory risks.” 

We are aware of cross-sectional studies that detected the significant association between e-cigarette use and respiratory diseases (e.g., asthma and COPD), and we cited those studies in the Introduction. As our study analyzed longitudinal epidemiological data (the PATH Study), we focused on discussing the (consistent and inconsistent) findings from prior studies based on PATH data in the Discussion section.

COMMENT: 2. Definition of variables: 2a. The definition of current smoker apparently is not current exclusive smoker; this category includes poly users. Unless I missed something, that is not exclusive cigarette smoking because it includes individuals who used multiple tobacco products. This needs to be made very clear. 2b. Similarly for e-cigarette use, is this really e-cigarette use exclusive of smoking? And of any other tobacco products? And what about marijuana? This is a significant issue because in many samples polyuse is the norm, and readers need to know exactly how the categories were defined. I don't know if the authors should be asked to reanalyze their data but I think this issue needs to highlighted in a revision.

RESPONSE: As described in the “Key measures” section, current smokers were defined as “past 30-day use of any combustible tobacco products”, and combustible tobacco products include cigarettes, traditional cigars, cigarillos, filtered cigars, pipes, and hookah. 

Regarding marijuana use, we have revised the Limitation section as “…There may be other variables (e.g., marijuana use) affecting the outcome. Future research is warranted to replicate the analyses accounting for those variables (e.g., cigarette pack years and marijuana use [15]).”

COMMENT: 3. The presentation of analyses and results is still confusing. When the authors say an analysis is on “important respiratory symptoms,” apparently this means a binary criterion variable. At other points, respiratory symptoms means a continuous index of symptoms. The Ms goes back and forth without justifying why one of these is important and the other isn’t. Apparently, they think this is intuitively obvious but it isn't obvious to me.

RESPONSE: We apologize for the confusion. We have revised the “Key measures” section that “When used as the outcome variable, it was coded as a binary variable and an index score of ≥2 was defined as having important respiratory symptoms [17].” 

As described in the “Statistical analysis” section, we examined the association between Wave 4 e-cigarette use status (current/former use vs. never use) and Wave 5 respiratory symptom index (≥2 vs. <2) by combustible tobacco smoking status, “adjusting for Wave 4 respiratory symptom index (as a continuous variable)” and other covariables.

In sum, as an outcome variable (Wave 5), the respiratory symptom index was coded as a binary variable. We controlled for Wave 4 respiratory symptom index (covariable) because baseline respiratory symptom index would affect follow-up respiratory symptom index, and as a covariable we used the original value (a continuous variable). 

COMMENT: 4. It is still unclear to me how the authors can say that their longitudinal analyses are based on participants without any respiratory disease at baseline, and then report the mean number of respiratory symptoms they had at baseline Maybe I’m dense (I’ve been accused of worse) but it’s difficult for me to understand how you can have it both ways.

RESPONSE: We have revised the sentences to clarify that respondents were those who “reported no diagnosis of any respiratory diseases”. It is likely that respondents had respiratory symptoms but had never been diagnosed with any respiratory disease. We controlled for baseline respiratory symptom index because baseline respiratory symptoms would affect follow-up respiratory symptoms, that is people with more respiratory symptoms at baseline were more likely to report functionally important respiratory symptoms at follow-up.

COMMENT: 5. The authors say that coughing is a measure of chronic airway obstruction as in asthma. I believe that coughing can also be attributable to infectious causes, i.e., upper respiratory infections such as colds or flu.

RESPONSE: We completely agree with the reviewer and therefore, have deleted the sentence in the Introduction section.

---

## [Editor Report · Decision Letter 2]

19 Feb 2024

Longitudinal association between e-cigarette use and respiratory symptoms among US adults: Findings from the Population Assessment of Tobacco and Health Study Waves 4 – 5

PONE-D-23-19648R2

Dear Dr. Jiang,

We’re pleased to inform you that your manuscript has been judged scientifically suitable for publication and will be formally accepted for publication once it meets all outstanding technical requirements.

Kind regards,

Sadaf G. Sepanlou, MD/MPH

Academic Editor

PLOS ONE
---

## [Editor Report · Acceptance letter]

20 Feb 2024

PONE-D-23-19648R2 

PLOS ONE

Dear Dr. Jiang, 

I'm pleased to inform you that your manuscript has been deemed suitable for publication in PLOS ONE. Congratulations! Your manuscript is now being handed over to our production team.

Kind regards, 

on behalf of

Dr. Sadaf G. Sepanlou 

Academic Editor

PLOS ONE